# The Fast Growth and Quick Spread of Synchronous Tumors

**DOI:** 10.3390/diagnostics13162706

**Published:** 2023-08-19

**Authors:** Li-Yu Chen, Yu-Hung Chen, Yen-Kung Chen

**Affiliations:** 1Department of Family Medicine, Shin Kong Wu Ho-Su Memorial Hospital, Taipei 111, Taiwan; leisurepika@gmail.com; 2Department of Nuclear Medicine and PET Center, Shin Kong Wu Ho-Su Memorial Hospital, Taipei 111, Taiwan; 3School of Medicine, Fu Jen Catholic University, New Taipei City 242, Taiwan

**Keywords:** synchronous tumor, buccal cancer, lung adenocarcinoma, FDG PET/CT

## Abstract

A 47-year-old man was diagnosed with left buccal squamous cell carcinoma using FDG PET/CT, by which focal lesions in the left buccal and left neck lymph node were found. Three months after the operation, CT images revealed a left lower lung lesion. Pathology indicated a left lower lung adenocarcinoma. Second FDG PET/CT was performed more than 11 days later, and lesions with intense FDG uptake were found in the left lower lung, metastatic to the lymph nodes, lungs, bones, and liver. The prior FDG PET/CT scan showed negative findings in the lungs. However, lung cancer with multiple metastases appeared 4 months later.

**Figure 1 diagnostics-13-02706-f001:**
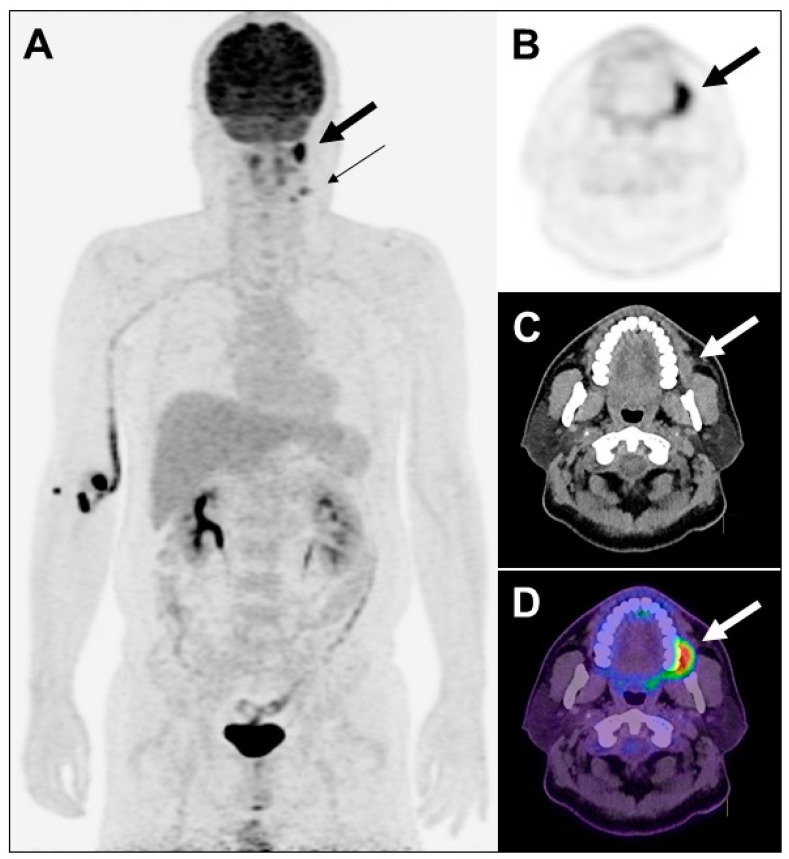
Maximum intensity projection view of PET (**A**), transaxial views of PET (**B**), CT (**C**) and PET/CT fusion (**D**) images of the illustrative case. A 47-year-old man with history of newly diagnosed left buccal poorly differentiated squamous cell carcinoma, who underwent PET/CT with ^18^F-fluorodeoxyglucose (FDG), was found to have a focal lesion (2.3 × 1.1 × 2 cm) with moderate increased FDG uptake ((**A**,**B**) SUV_max_ 5.5) in the left buccal ((**A**–**D**) thick arrow). There were two foci with mild increased FDG uptake (SUV_max_ 2.3 and 1.3) in the left neck level I ((**A**) 0.94 cm; thin arrow) and level II (0.8 cm), respectively. Bilateral lungs were clear and no lesion was detected. Image staging was T2N1M0. Four days later, he received wide excision for a left buccal tumor and left modified radial neck dissection. Pathology reported a verrucous and ulcerated tumor measuring 4.1 × 2.7 × 1.3 cm. The modified radical neck dissection samples showed left neck level I one out of five metastatic squamous cell carcinoma, and left neck level II one out of one negative for malignancy. Pathologic staging was pT3N1. Later, he received concurrent chemoradiotherapy with 66Gy/33Fx plus concurrent cisplatin 6 cycles. Patients with a large metabolic tumor volume on FDG-PET may experience poor clinical courses [1].

**Figure 2 diagnostics-13-02706-f002:**
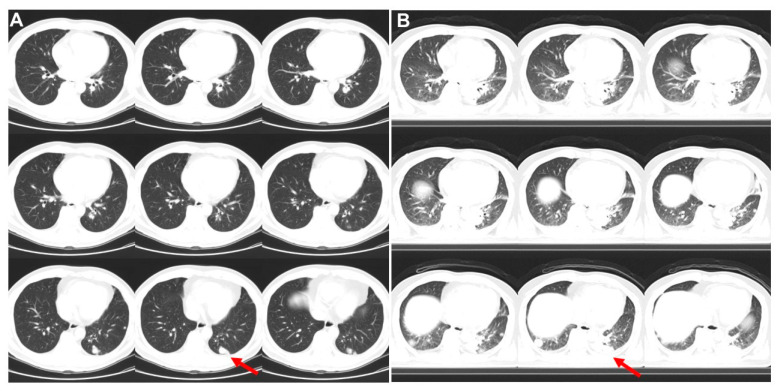
Three months and 21 days later, a series of transverse CT images (**A**) from left to right and up to down revealed the left lower lung lesion (1.1 cm; arrow) and contact to pleura on lung window. A small node close to aorta was also noted. Besides, small nodular lesions involving the mediastinal, lung and liver were detected. He received biopsy and histopathological study showed left lower lung adenocarcinoma. A further 11 days later, 2nd FDG PET/CT was performed. CT images (**B**) revealed the left lower lung lesion was enlarged (3.1 cm; arrow) and spread to aorta margin and pleura. Bilateral lung nodules also became enlarged. To get an idea about how fast lung cancer grows, it’s helpful to look at doubling time [2,3]. Since the left low lung lesion grew from 1.1 × 1.1 × 0.7 cm by CT scan to 3.1 × 1.9 × 1.5 cm by 2nd PET/CT scan, its doubling time was 4 days.

**Figure 3 diagnostics-13-02706-f003:**
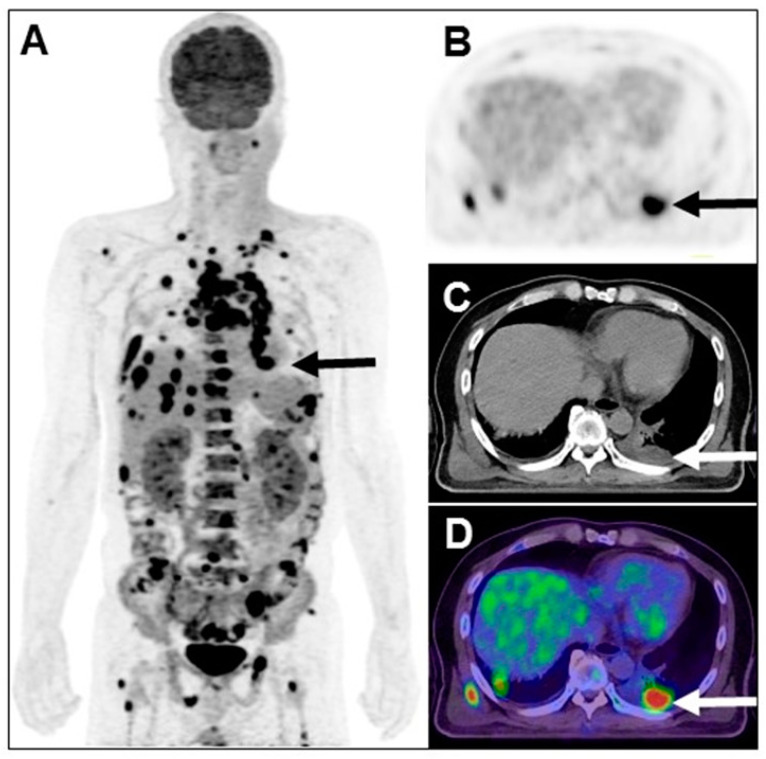
The 2nd FDG PET/CT revealed intense FDG uptake in the left low lung ((**A**,**B**) SUV_max_ 13.6; arrow) contact to pleura ((**C**,**D**) 3.4 cm; arrow), metastatic to the mediastinal nodes, bilateral supraclavicular nodes, bilateral lungs, right subscapular muscle, bones and liver (**A**). All metastatic lesions showed intense FDG uptake, indicating that FDG metabolic status was the same as the left lower lung adenocarcinoma. Several studies have demonstrated that cancer patients, compared to the general population, had a higher risk of developing new primary tumors [4,5,6]. The series of neutrophil-to-lymphocyte ratio progressed with the following trend: from 39.6/45.7 to 69/16 when received concurrent chemoradiotherapy, increasing to 86.4/3.6 during the 2nd tumor of the left lower lung adenocarcinoma biopsy, to 91.5/3 in the 2nd FDG PET/CT, and finally to 96/1. In this case, the second lung adenocarcinoma following buccal squamous cell carcinoma developed much faster and more aggressive than in previous studies and progressed to higher neutrophil-to-lymphocyte ratio. Patient expired 11 days after the 2nd PET/CT examination. Early appearance of abnormal neutrophil-to-lymphocyte ratio associated with further examination may indicate the rapid growth and metastasis of cancer, and hence offers diagnostic and management opportunities [7,8].

## Data Availability

Date available on request from the authors.

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
