# Peer review of "The Fast Growth and Quick Spread of Synchronous Tumors"

_diagnostics, 2023, doi:10.3390/diagnostics13162706_

Round 1

Reviewer 1 Report

The concept of the study is interesting, so I recommend it to be published in Diagnostics.

no 

Author Response

Thanks

Reviewer 2 Report

I think the find is quite interesting. For this reason I believe that a case report can be compiled, rather than a simple review of images. In this way, the case could be described more fully and a review of the literature carried out. This would result in more recent literature review focusing on the occurrence of multiple tumors and the growth and spread rate of lung adenocarcinoma

Author Response

Thanks.

The cause of synchronous tumors may be due to immunity or others.  Then undetermined the pathogenesis. So, interesting image only.

Reviewer 3 Report

This paper explores an interesting case of buccal SCC surgery followed by rapid lung cancer development, hypothesizing a causal link.

Strength: Unique clinical scenario that may shed light on unexplored connections between the two cancers.

Question: Causal link between surgery and lung cancer needs further discussion; do the author think the lung cancer will not occure if surgery for buccal SCC was not performed?

Recommendation: Since this paper is related to FDG and its abilitiy of measuring malignant potential, it would be great if you can cite the following paper: Uchiyama Y, Hirata K, Watanabe S, et al. Development and validation of a prediction model based on the organ-based metabolic tumor volume on FDG-PET in patients with differentiated thyroid carcinoma. Ann Nucl Med. 2021 Nov;35(11):1223-1231.

Author Response

Question: Causal link between surgery and lung cancer needs further discussion; do the author think the lung cancer will not occure if surgery for buccal SCC was not performed?

Answer: Unknown. However, patient has risks factor for buccal cancer and lung cancer.

Recommendation: Since this paper is related to FDG and its abilitiy of measuring malignant potential, it would be great if you can cite the following paper: Uchiyama Y, Hirata K, Watanabe S, et al. Development and validation of a prediction model based on the organ-based metabolic tumor volume on FDG-PET in patients with differentiated thyroid carcinoma. Ann Nucl Med. 2021 Nov;35(11):1223-1231.

Answer: cite and described.

Round 2

Reviewer 2 Report

I respect the opinion of the authors, but I insist that the interest of the subject requires greater completeness in the text